# *RsSOS1* Responding to Salt Stress Might Be Involved in Regulating Salt Tolerance by Maintaining Na⁺ Homeostasis in Radish (*Raphanus sativus* L.)

**Wanting Zhang** [†], **Jingxue Li** [†], **Junhui Dong** [†], **Yan Wang, Liang Xu, Kexin Li, Xiaofang Yi, Yuelin Zhu** *[ORCID] and **Liwang Liu** *[ORCID]

National Key Laboratory of Crop Genetics and Germplasm Enhancement, Key Laboratory of Biology and Germplasm Enhancement of Horticultural Crop (East China), Ministry of Agriculture and Rural Affairs, College of Horticulture, Nanjing Agricultural University, Nanjing 210095, China; 2018104061@njau.edu.cn (W.Z.); 2020104062@stu.njau.edu.cn (J.L.); 2019204029@njau.edu.cn (J.D.); wangyanhs@njau.edu.cn (Y.W.); nauxuliang@njau.edu.cn (L.X.); likexin130132@163.com (K.L.); yixiaofang7091@163.com (X.Y.)
* Correspondence: ylzhu@njau.edu.cn (Y.Z.); nauliulw@njau.edu.cn (L.L.)
† These authors contributed equally to this work.

**Abstract:** Radish is a kind of moderately salt-sensitive vegetable. Salt stress seriously decreases the yield and quality of radish. The plasma membrane Na⁺/H⁺ antiporter protein Salt Overly Sensitive 1 (SOS1) plays a crucial role in protecting plant cells against salt stress, but the biological function of the *RsSOS1* gene in radish remains to be elucidated. In this study, the *RsSOS1* gene was isolated from radish genotype 'NAU-TR17', and contains an open reading frame of 3414 bp encoding 1137 amino acids. Phylogenetic analysis showed that RsSOS1 had a high homology with BnSOS1, and clustered together with *Arabidopsis* plasma membrane Na⁺/H⁺ antiporter (AtNHX7). The result of subcellular localization indicated that the RsSOS1 was localized in the plasma membrane. Furthermore, *RsSOS1* was strongly induced in roots of radish under 150 mmol/L NaCl treatment, and its expression level in salt-tolerant genotypes was significantly higher than that in salt-sensitive ones. In addition, overexpression of *RsSOS1* in Arabidopsis could significantly improve the salt tolerance of transgenic plants. Meanwhile, the transformation of *RsSOS1△999* could rescue Na⁺ efflux function of AXT3 yeast. In summary, the plasma membrane Na⁺/H⁺ antiporter RsSOS1 plays a vital role in regulating salt-tolerance of radish by controlling Na⁺ homeostasis. These results provided useful information for further functional characterization of RsSOS1 and facilitate clarifying the molecular mechanism underlying salt stress response in radish.

**Keywords:** radish; *RsSOS1*; salt stress; RT-qPCR; overexpression; yeast functional complement

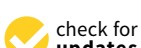

## 1. Introduction

Radish (*Raphanus sativus* L.), an essential root vegetable crop of the Brassicaceae family, playing an important role in vegetable production and supply. It has rich nutrition and high medicinal value and is very popular worldwide, especially in East Asia. Salt stress has become one of the major destructive abiotic stresses affecting crop growth, resulting in the decline of crop yield and quality [1]. It is predicted that by 2050, nearly half of the world's cultivated land will be salinized, posing a great threat to sustainable agricultural development and crop safety production [2]. Soil salinization and secondary salinization giving rise to salt stress could seriously affect the yield and quality of radish taproot. Therefore, the exploration and functional analysis of salt-tolerance-related genes in radish will provide a solid theoretical basis for further analyzing the mechanism underlying radish response to salt stress and developing salt-tolerant germplasm.

Plants display resistance to salt stress in several ways, including osmotic regulation, ion homeostasis regulation, hormone regulation, scavenging of reactive oxygen species

(ROS) and signal transduction [3]. Reducing the content of Na$^+$ in the cytoplasm of plants is a vital way to maintain intracellular ion homeostasis under salt stress, mainly including reduction of Na$^+$ absorption, promotion of Na$^+$ efflux or Na$^+$ compartmentation [4]. Among them, exclusion and compartmentation of Na$^+$ are executed by Na$^+$/H$^+$ antiporter proteins located on the plasma and vacuole membrane, respectively. The gene of the vacuolar membrane Na$^+$/H$^+$ exchanger, *RsNHX2*, has been identified, which plays a key role in resistance to salt stress in radish [5]. The role of Salt Overly Sensitive 1 (SOS1), a plasma membrane Na$^+$/H$^+$ antiporter, has been widely explored in salt tolerance. It eliminates Na$^+$ from cells under the action of plasma membrane H$^+$-ATPase to reduce the accumulation of Na$^+$ in cells [6,7]. *SOS1* homologues have been isolated in a variety of higher plant species, such as *Sesuvium portulacastrum* [8], Rice [9], *Karelinia caspia* [10], etc., while the isolation of the *RsSOS1* gene and its functional response to salt stress in radish have not been reported.

In Arabidopsis, the SOS1 protein contains three conserved functional domains necessary to regulate its protein activity, namely Nhap, InhiBD and S2P under normal conditions, the inhibited region 'InhiBD' at the C-terminal of SOS1 combines with the adjacent functional active region 'Nhap' to inhibit its activity, and the Na$^+$ content in cells remains stable. When subjected to salt stress, the SOS signal transduction pathway is activated. Under this condition, the SOS2-SOS3 protein kinase complex phosphorylates the 1136th and 1138th serine sites of SOS1 S2P, which releases and activates the ion transport functional domain from the self-inhibitory state and plays the function of Na$^+$/H$^+$ exchange transport under the action of H$^+$ ATPase [11]. However, the functional domain in RsSOS1 has not been explored.

Based on the previous transcriptome data, it was found that the *RsSOS1* gene was significantly upregulated under salt stress [12], implying that *RsSOS1* might play a significant role in the stress response to salt in radish. In this study, to explore molecular characteristics and function of the *SOS1* gene in radish, the *RsSOS1* gene was isolated and subcellular localization was analyzed. Additionally, the expression profiles of the *RsSOS1* gene in different salt-tolerant radish genotypes under salt stress were investigated with real-time quantitative PCR. Furthermore, the biological function of *RsSOS1* in salt stress response was investigated by overexpression in *Arabidopsis* and functional complement of yeast. The results of this study may facilitate elucidating the molecular mechanism of response to salt stress and improving the salt-tolerant traits in radish genetically.

## 2. Materials and Methods

### 2.1. Isolation of RsSOS1 Gene in Radish

Total RNA was extracted from the root tissue of radish advanced inbred line 'NAU-TR17' using the RNA Simple Total RNA Kit (Tiangen biotech Ltd. Co., Beijing, China). The cDNA was synthesized using PrimeScript RT reagent Kit (TaKaRa, Dalian, China). The *RsSOS1* gene was isolated with the cDNA template. The primer *RsSOS1-F/R* was designed using homologous recombination primer-design software CEDesign (Vazyme Biotechnology Co., Ltd., Nanjing, China) according to the CDS sequence of *RsSOS1* (XM_018601043.1) in NCBI (Table S1). The target fragment was combined with pCAMBIA1300 vector (named 35S::RsSOS1-GFP) by the homologous recombination method, and then transferred into *E.coli* DH5α (TransGen Biotechnology Co., Ltd., Beijing, China) with the heat shock method [13]. The positive clones identified by PCR were sequenced (Springen Biotechnology Co., Ltd., Nanjing, China). The gene-specific primers for isolating *RsSOS1* genomic DNA were designed according to the gDNA sequence of *RsSOS1* in genomic database (Table S1) [14]. The PCR products were recovered and purified, and after the T-A cloning, the positive clones were then sequenced to obtain the full-length sequence of *RsSOS1* gDNA.

## 2.2. Bioinformatic Analysis of RsSOS1

The CDS sequence of the *RsSOS1* gene sequencing was translated into protein. The transmembrane domain (http://www.cbs.dtu.dk/services/TMHMM/, accessed on 1 April 2021), physical and chemical properties (https://web.expasy.org/protparam/, accessed on 1 April 2021) and signal peptides (http://www.cbs.dtu.dk/services/SignalP-5.0/, accessed on 1 April 2021) of RsSOS1 were analyzed. To compare the genetic relationship of SOS1 from different species and identify the membrane localization of RsSOS1, a phylogenetic tree of 12 SOS1 proteins from different species coupled with eight different membrane located *Arabidopsis* $Na^+/H^+$ antiporter proteins was mapped with Mega 6.0 software (Mega Limited, Auckland, New Zealand). It has been reported that AtNHX1-AtNHX6 were located on cell endomembrane, while AtNHX7 and AtNHX8 were located on plasma membrane [15,16]. An unrooted phylogenetic tree was generated subsequently with the neighbor-joining (NJ) method with: p-distance (Model/Method), pairwise deletion (Gaps/Missing Data treatment) and 1000 bootstrap replicates (Test of phylogeny) [17,18].

## 2.3. Subcellular Localization of RsSOS1 Protein

The Agrobacterium GV3101, transferred with expression vector 35S::RsSOS1-GFP by the freezing–thawing method, was injected into tobacco leaves for transient transformation [19]. The subcellular localization of 35S::RsSOS1-GFP in tobacco cells was observed under laser confocal microscope (LSM 800, Zeiss, Jena, Germany).

## 2.4. Identification of Salt Tolerance of Radish Genotypes

Using the radish advanced inbred lines, 'NAU-TR17' and 'NAU-TR12' as materials, plants with the same growth status were selected and hydroponic stress treatment was conducted with 250 mmol/L NaCl contained nutrient solution [20], and nutrient solution without NaCl was used as control. After treatment for 10 d, the relative values of growth indexes including relative plant height, root length, shoot fresh weight, root fresh weight and total biomass were calculated after measurement. The calculation was as follows: Relative value = measured value in the treatment group/measured value in the control group × 100%; Biomass (fresh weight/g) = shoot fresh weight + root fresh weight.

## 2.5. Expression Profiling of RsSOS1

Radish genotypes 'NAU-TR17' and 'NAU-TR12' were used as materials. Salt stress treatment with 150 mmol/L NaCl [13] was performed at seedling stage, and the non-NaCl treated material was used as control. The roots and leaves were collected at 0 h, 3 h, 6 h, 12 h, 24 h, 48 h and 72 h after treatment and three biological replicates were performed. Total RNA was extracted using the RNA Simple Total RNA Kit (Tiangen biotech Ltd. Co., Beijing, China). The cDNA was synthesized using PrimeScript RT reagent Kit (TaKaRa, Dalian, China). Primer-Blast program (https://www.ncbi.nlm.nih.gov/, accessed on 1 April 2021) was used to design real-time quantitative PCR (RT-qPCR) primers (*RsSOS1-qRT-F/R, RsActinq-RT-F/R*) (Table S1), and *RsActin* gene was used as an internal reference gene [21]. RT-qPCR was performed according to the instructions of SYBR Premix Ex Taq kit (TaKaRa, Dalian, China) and the reaction procedures were 95 °C for 10 min, 95 °C for 10 s, 56 °C for 10 s, 72 °C for 20 s and 50 °C for 30 s, 40 cycles were set from step 2 to step 4. The expression level was estimated using the $2^{-\triangle\triangle CT}$ method [22].

## 2.6. Heterologous Expression of RsSOS1 in Arabidopsis and Yeast

Engineered *Agrobacterium* carrying 35S::RsSOS1-GFP were activated and used to infect the inflorescence of *Arabidopsis* [23]. The positive transgenic plants were screened continuously on 1/2 Murashige Skoog medium containing 36 mg/L hygromycin to obtain the homozygous $T_3$ generation plants. Using DNA of wildtype (WT) and transgenic *A. thaliana* as templates, the transgenic positive plants were identified by amplification with pCAMBIA1300 vector universal primer. The expression of *RsSOS1* was analyzed by RT-PCR. The wildtype and transgenic *Arabidopsis* plants were sown on 1/2 MS medium

containing 0, 75 and 150 mmol/L NaCl, respectively, and the germination rate, root length and fresh weight (Fw) were counted and calculated.

To analyze the sodium transport function of RsSOS1 protein, AXT3 yeasts (endogenous vacuolar membrane Na$^+$ transporter NHX1, plasma membrane Na$^+$ transporter ENA1–4 and NHA1-deficient: *nhx1::TRP1, ean1::HIS3::ena4, nha1::LEU2*) were used for functional complement analysis (kindly provided by Professor Yu L., College of Resources and Environmental Sciences, Nanjing Agricultural University). The alignment was carried out between amino acid sequences of RsSOS1 and AtSOS1 to the conjectured self-activity inhibited region of RsSOS1. It was speculated that the self-activity inhibited region of RsSOS1 was located behind the 999th amino acid according to the known self-activity inhibited region of AtSOS1. The cloning primers of *RsSOS1* and *RsSOS1△999* were designed by the homologous recombination method, respectively (*RsSOS1-JM-F/R, RsSOS1△999-JM-F/R*) (Table S1). The sequence of *RsSOS1△999*, which has been removed from the C-terminal inhibited region of *RsSOS1*, contains first 2997 bp nucleotide sequence of *RsSOS1* plus the stop codon. The clone product of *RsSOS1* and *RsSOS1△999* were combined with yeast expression vector *pYES2* (http://www.youbio.cn, accessed on 23 October 2021), respectively.

The *pYES2* plasmid, recombinant plasmid *pYES2-RsSOS1* and *pYES2-RsSOS1△999* were transformed into AXT3 yeast by the lithium acetate method, respectively. These three kinds of transformed yeast were incubated overnight in SD-Ura liquid medium (Coolaber Technology Co., Ltd., Beijing, China), and then resuspended with sterile water after centrifugation, respectively. In total, 6 μL yeast liquid was added to the SC-Ura+Gal solid medium with concentration of 0, 35, 75, 150, 300 mmol/L NaCl after diluting these three kinds of yeast liquid according to the proportion of 1, $10^{-1}$, $10^{-2}$, $10^{-3}$, $10^{-4}$, respectively. Then, the growth of yeast was observed [24]. The yeast were collected and baked at 80 °C for 72 h after cultivating these three kinds of yeast for 48 h in SC-Ura+Gal liquid medium containing 0 and 75 mmol/L NaCl, respectively. Then, 0.1 g samples were dissolved to determine the contents of potassium and sodium ions by plasma emission spectrometer (iCAP7000 plus, Thermo Fisher, Waltham, MA, USA) [25].

### 2.7. Statistical Analysis

The experimental data were analyzed using SPPS (SPSS Inc., Chicago, IL, USA). Significance tests for differences between control and stress treatments were assessed at a $p \leq 0.05$ level of significance. All experiments were performed and analyzed separately based on three biological replicates.

## 3. Results

### 3.1. Isolation and Phylogenetic Analysis of RsSOS1 Gene

The cDNA coding sequence of the *RsSOS1* gene with a length of 3414 bp was isolated with the cDNA from 'NAU-TR17', and the corresponding gDNA sequence of *RsSOS1* is 6113 bp in length and contains 23 exons (Figure S1). These sequences were deposited in GenBank with the Acc. No. MZ484951 and No. MZ484950, respectively. Sequence analysis suggested that *RsSOS1* encoded 1137 amino acids with a relative molecular weight of 125.61 kDa and an isoelectric point of 6.61. The N-terminal region contains 12 transmembrane domains, with a long hydrophilic tail in the C-terminal part (Figure S2). Moreover, the signal peptide does not exist in this protein.

Phylogenetic analysis showed that RsSOS1 has a high homology with BnSOS1 (Figure 1), and is closely related to the plasma membrane Na$^+$/H$^+$ antiporter of *Arabidopsis* (AtNHX7, AtNHX8) in evolutionary relationship, while is far less related to the vacuolar Na$^+$/H$^+$ antiporters of *Arabidopsis* (AtNHX1–6), suggesting that RsSOS1 is a plasma membrane-located protein.

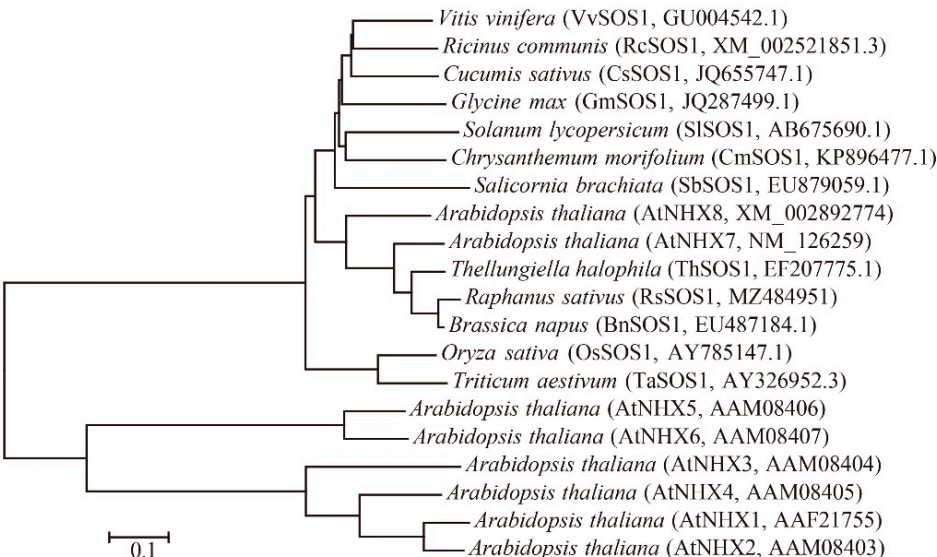

**Figure 1.** Phylogenetic analysis of Na$^+$/H$^+$ antiporters in different species.

### 3.2. Subcellular Localization Analysis of RsSOS1

The subcellular localization of RsSOS1 protein in tobacco leaves was observed under laser confocal microscope (Figure 2). The results indicated that the green fluorescence signal of 35S::RsSOS1-GFP was concentrated on the cell membrane, indicating that the RsSOS1 of radish plays corresponding functions in the cell membrane.

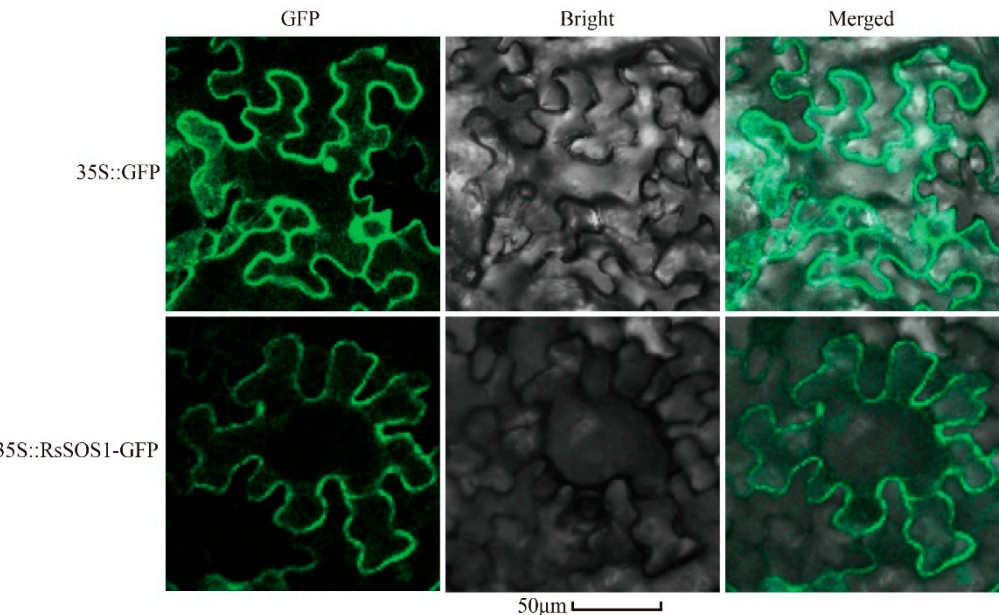

**Figure 2.** The subcellular localization of RsSOS1 in tobacco cells.

### 3.3. Identification of Salt Tolerance of Genotype 'NAU-TR17' and 'NAU-TR12'

The growth and development of genotype 'NAU-TR17' were normal after salt treatment (250 Mm NaCl), with no obvious yellowing leaves and normal of fibrous roots compared with the control plants. However, the genotype 'NAU-TR12' suffered serious damage after salt treatment, with retarded fibrous root growth and yellowing wilted leaves (Figure 3A).

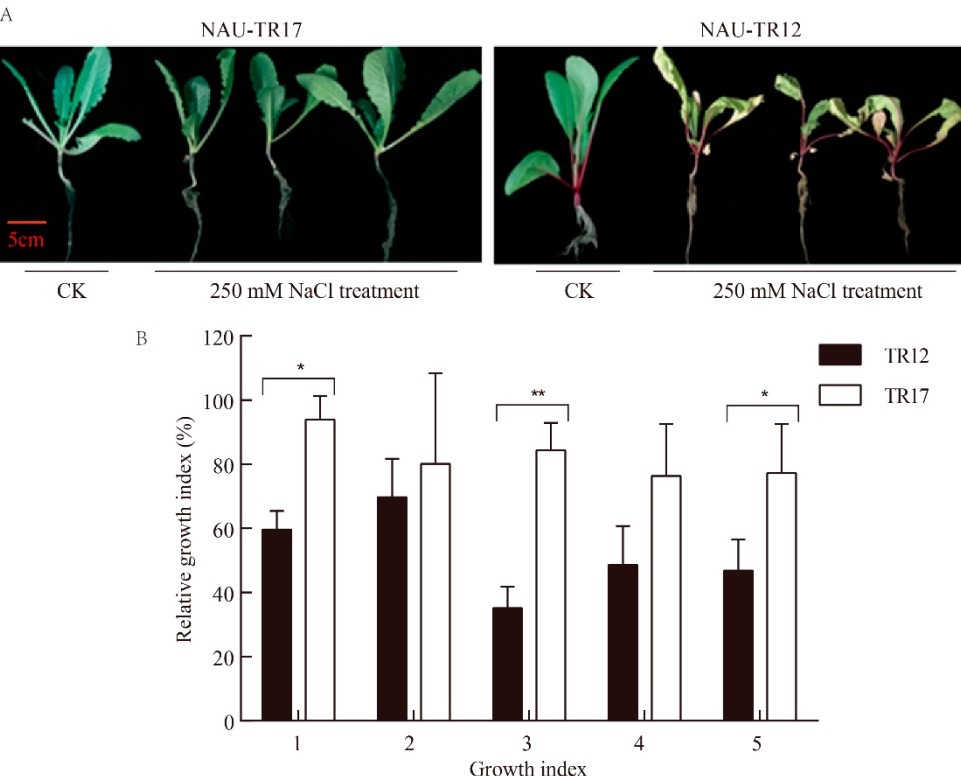

**Figure 3.** Salt tolerance evaluation of radish genotypes: (**A**) Phenotypic identification of salt tolerance; (**B**) Growth indexes under 250 mmol/L NaCl treatment. 1–5 represents relative plant height, root length, fresh weight of shoot, fresh weight of root and biomass, respectively. * and ** means differ significantly ($p < 0.05$ and $<0.01$, respectively).

The relative plant height, the relative underground fresh weight and the relative biomass of 'NAU-TR17' remained at a high level under 250 mmol/L NaCl condition. However, the relative value of 'NAU-TR12' decreased seriously, and the plant growth was seriously inhibited (Figure 3B). It could be concluded that genotype 'NAU-TR17' has strong salt tolerance, while genotype 'NAU-TR12' is sensitive to salt stress.

### 3.4. Expression Profiling of RsSOS1 under Salt Stress

The expression level of the *RsSOS1* gene in the salt-tolerant genotype 'NAU-TR17' root was upregulated after 6–72 h of salt treatment (Figure 4). In the leaves of 'NAU-TR17', there was no significant difference in expression level within 12 h after salt treatment, and the expression level was downregulated after 12 h. In the salt-sensitive genotype 'NAU-TR12' root, the expression level of *RsSOS1* was significantly upregulated only at 12 h to 24 h after salt treatment, while the expression difference was not significant during other treatment. In 'NAU-TR12' leaves, the expression level of *RsSOS1* was upregulated at 48 h after salt treatment, while the expression difference was not significant during other treatments. The significance of differential expression level of *RsSOS1* gene in roots of salt sensitive genotype 'NAU TR12' was far less than that in salt tolerant genotype 'NAU TR17' under salt stress, implying that the *RsSOS1* gene contributed greatly to the salt tolerance of radish.

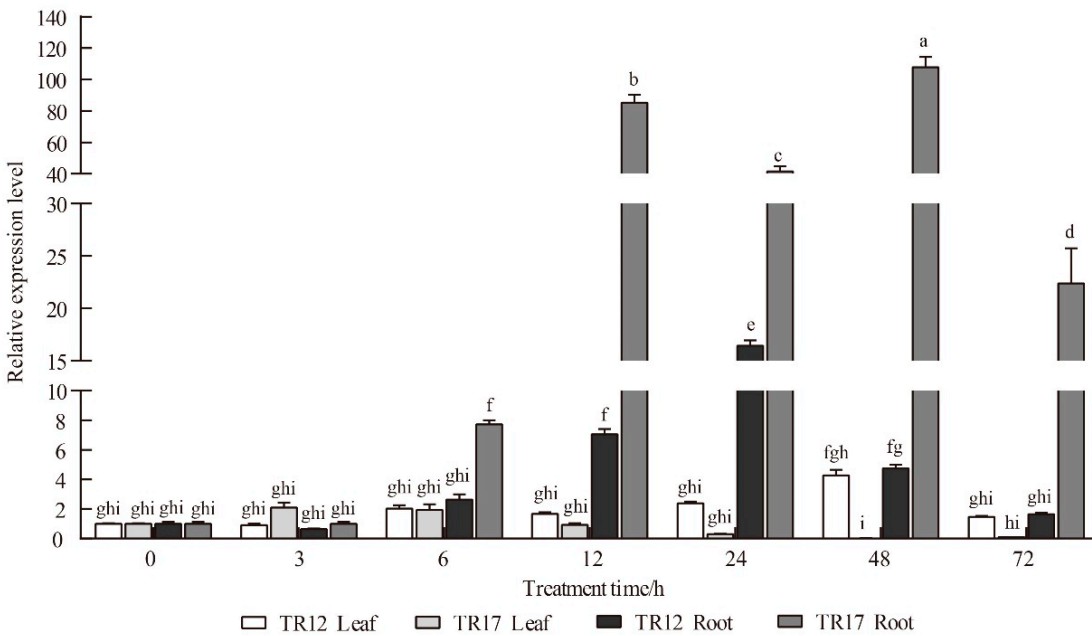

**Figure 4.** Expression profiles of *RsSOS1* in radish leaf and root under salt stress. Error bars indicate SD, letters represent significant differences at *p* < 0.05 level based on Duncan's test.

### 3.5. Overexpression of RsSOS1 in Arabidopsis to Improve Salt Stress Tolerance

PCR amplification results showed that the *RsSOS1* was successfully transferred into *Arabidopsis* (Figure 5A). RT-PCR results indicated that *RsSOS1* could be expressed in transgenic plants (Figure 5B). The germination rates of three transgenic *Arabidopsis* lines with different overexpression levels (OE1-OE3) and wildtype (WT) were compared under 0, 75 and 150 mmol/L NaCl treatment. There was no significant difference in germination rate between WT and OE1-OE3 in the control medium, while the germination rate was totally decreased under NaCl stress at 75 mmol/L and 150 mmol/L (Figure 5C). The germination rates of WT and OE1-OE3 under NaCl treatment at 75 mmol/L were 82.82% and 90.71–94.29%, respectively. Under NaCl treatment at 150 mmol/L, the germination rates of OE1-OE3 were 82.14–84.52% but higher than that of control (71.42%) (Figure 5D). It could be concluded that overexpression of *RsSOS1* gene could increase the seed germination rates under salt stress.

When compared with transgenic plants, the WT plants showed more wilted and yellow leaves under NaCl condition at 150 mmol/L (Figure 5E). The growth of WT and transgenic plants was inhibited under NaCl treatment at 75 mmol/L, but the fresh weight of OE-*RsSOS1* transgenic plants was higher than that of WT, and root length was longer than that of WT (Figure 5F,G), indicating that the overexpression of *RsSOS1* gene could improve the salt tolerance of transgenic plants.

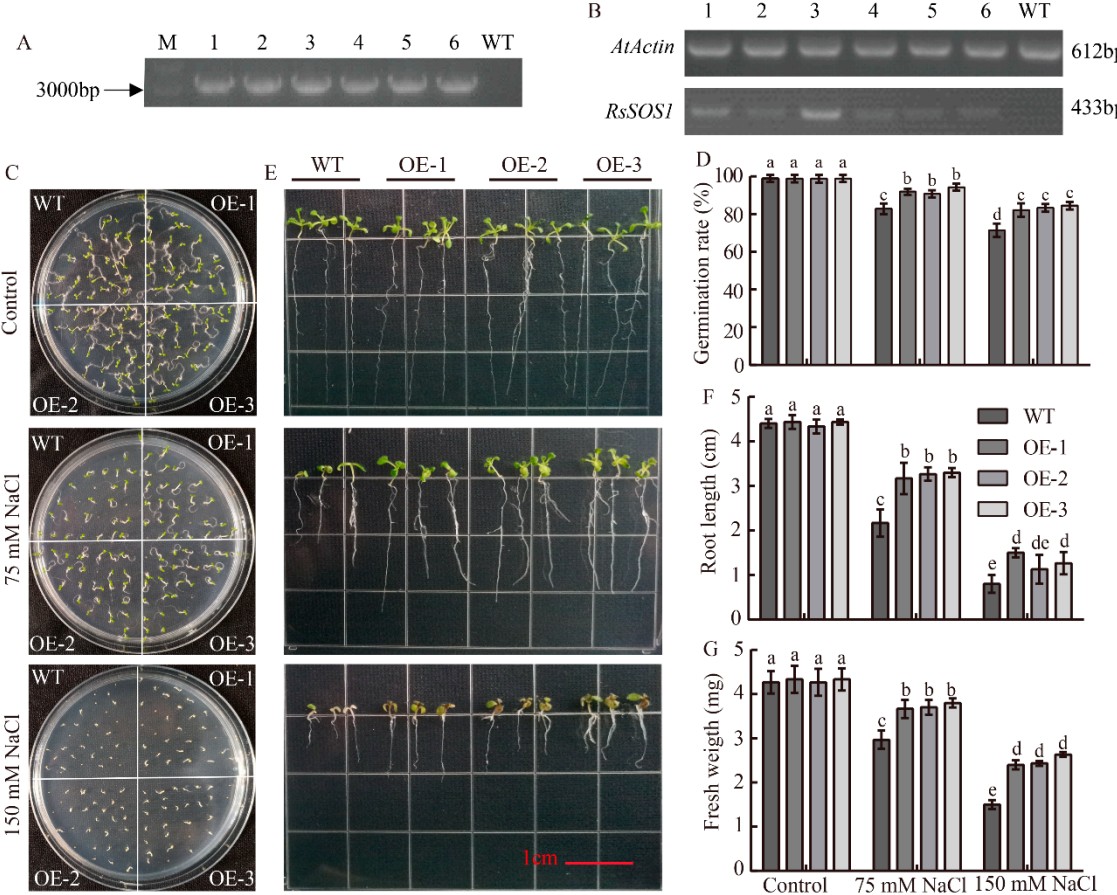

**Figure 5.** Overexpression of *RsSOS1* in Arabidopsis. (**A,B**): PCR identification and RT-PCR validation of *RsSOS1* transgenic Arabidopsis, respectively; (**C**) The germination of OE-*RsSOS1* transgenic *Arabidopsis* under salt stress; (**D**) Statistics of germination rate of OE-*RsSOS1* plant under salt stress; (**E**) Phenotype of OE-*RsSOS1* under NaCl stress; (**F,G**) Statistics of root length and fresh weight. M, DL 5000 Marker. WT, wild type *Arabidopsis*. 1–6, six transgenic *Arabidopsis* lines. Error bars indicate SD, letters represent significant differences at *p* < 0.05 level based on Duncan's test.

### 3.6. Functional Complement Analysis of RsSOS1 Protein in AXT3 Yeast

On the medium with the supplement of 0 and 35 mmol/L NaCl, there was no significant difference among the growth of *pYES2, RsSOS1* and *RsSOS1△999* transformed yeast (Figure 6). Under the conditions of 75, 150 or 300 mmol/L NaCl treatment, there was no significant difference between the growth of *pYES2* and *RsSOS1* transformed yeast, and the growth became weaker with the increase of salt concentration. When the salt concentration reached 300 mmol/L, these two yeast transformers could hardly grow. With the increase of salt concentration, the growth of *RsSOS1△999* transformed yeast was significantly better than that of *pYES2* and *RsSOS1* transformed yeast, and it could grow well when the concentration of NaCl was up to 300 mmol/L. These findings indicate that transfection of *RsSOS1△999* could restore the growth of the dysfunctional yeast mutant AXT3 under high salt conditions.

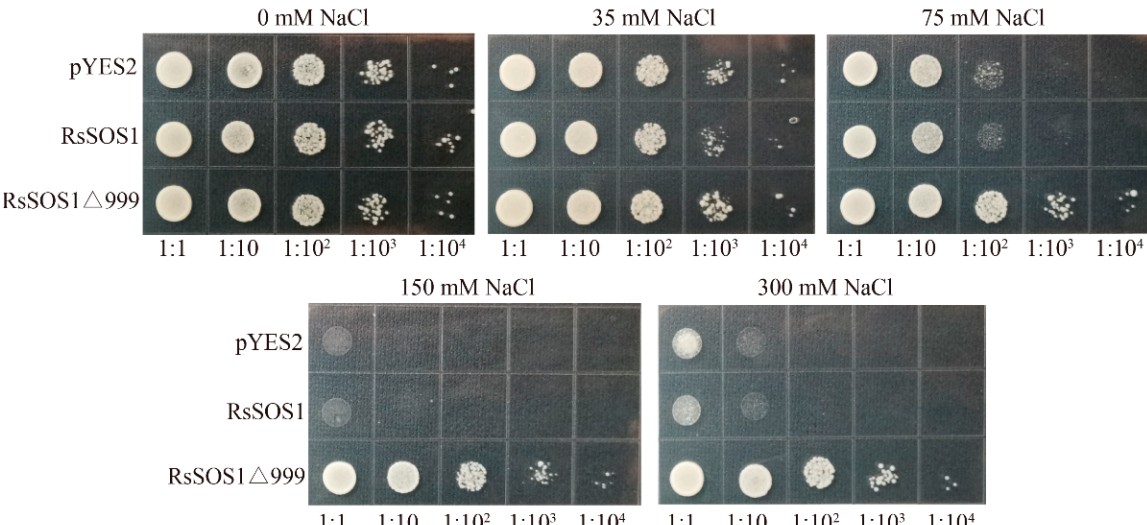

**Figure 6.** Function complements of RsSOS1 in AXT3 yeast. pYES2, AXT3 yeast transformed with blank vector *pYES2*. RsSOS1, Transgenic yeast with full length of *RsSOS1* gene. RsSOS1△999, Transgenic yeast with first 2997 bp of *RsSOS1* gene. Each kind of yeast transformant was diluted in multiples of 1, 10, $10^2$, $10^3$, $10^4$.

### 3.7. Determination of Ion Content in Transgenic Yeast

There was no significant difference in the content of $Na^+$ or $K^+$ among *RsSOS1*, *RsSOS1△999* and *pYES2* transformed yeast under normal conditions. However, when the content of $Na^+$ in yeast increased, $K^+$ content decreased under 75 mmol/L NaCl treatment (Figure 7). Under 75 mmol/L NaCl treatment, there was no significant difference in $K^+$ or $Na^+$ contents between transformers pYES2 and RsSOS1, while RsSOS1△999 strains had higher $K^+$ content, lower $Na^+$ content and higher $K^+/Na^+$ ratio, indicating that the first 999 amino acids of RsSOS1 contain $Na^+$ efflux functional structures, which played a critical role in maintaining intracellular ion homeostasis.

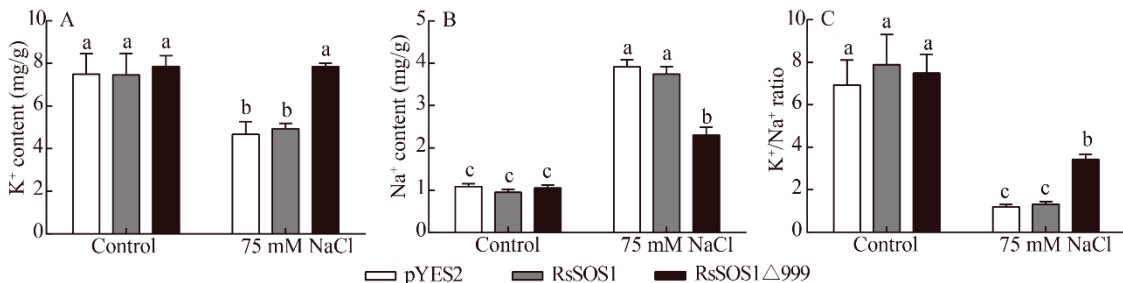

**Figure 7.** $K^+/Na^+$ ion content in transgenic yeast: (**A**) $Na^+$ content; (**B**) $Na^+$ content; (**C**). $K^+/Na^+$ ratio. Error bars indicate SD, letters represent significant differences at $p < 0.05$ level based on Duncan's test.

## 4. Discussion

The $Na^+/H^+$ antiporter encoded by the *SOS1* gene enables plants to adapt to the saline environment [26]. The ORF length of *SOS1* genes in most plant is 3410–3500 bp, encoding 1129–1169 amino acids, with a relative molecular weight of about 127 kDa and 10–12 transmembrane domains [27]. The results of *RsSOS1* sequence analysis in this study are consistent with those of previous studies. Subcellular localization indicated that RsSOS1 was localized on the cell membrane. Phylogenetic tree analysis suggested that RsSOS1 was closely related to plasma membrane $Na^+/H^+$ antiporters in evolutionary relationship, but far less related to inner membrane $Na^+/H^+$ antiporters, suggesting that RsSOS1 was a plasma membrane $Na^+/H^+$ antiporter, and played a corresponding role of maintaining $Na^+/K^+$ ion homeostasis in cell membrane. Previous studies have indicated that *OsSOS1*

was highly induced in the root after 15 h of salt stress treatment in rice [28]. In tuber mustard, both the expression levels of *BjSOS1-1* and *BjSOS1-2* were significantly induced in the root by salt treatment at 12 h [29], and the expression induction of *PabSOS1* was also found in *Populus* [30]. It was also found that *SOS1* was mainly induced in the root under salt stress in *Medicago sativa* [31] and *Cucurbita maxima* [32]. In this study, the *RsSOS1* gene was highly induced in root after 6 h of salt stress treatment in the salt-tolerant genotype 'NAU-TR17' taproot, which is in accordance with the reported results [28–32]. However, the *SOS1* gene of *Populus* and *Mesembryanthemum* is mainly upregulated in leaves [33,34]. Those results indicated that the functions of *SOS1* gene are relatively conserved and have various organ expression specificity among different species. In addition, the expression level of the *RsSOS1* gene in roots of the salt-sensitive genotype 'NAU-TR12' was far less than that in the salt-tolerant genotype 'NAU-TR17' under salt stress. It is speculated that the higher *RsSOS1* gene expression level of 'NAU-TR17' radish (salt tolerant) makes its root cells have a stronger ability to excrete sodium ions, so as to reduce the toxic effect of sodium ion accumulation in cells, which suggests that the expression level of the *RsSOS1* gene may be mainly responsible for the difference in salt tolerance among radish genotypes.

Since the first identification of the $Na^+/H^+$ antiporter gene *SOS1* in *A. thaliana* [35], the function of the *SOS1* gene in several plants has been investigated. Overexpression of the *TrSOS1* gene in Tamarix can supplement salt sensitivity of *sos1* mutants [36]. In addition, overexpression of cotton *GhSOS1*, soybean *GmSOS1* or *S. portulacastrum SpSOS1* in *Arabidopsis* could improve salt tolerance in transgenic plants [37–39]. In this study, through overexpression of the *RsSOS1* gene in *Arabidopsis*, it was found that transgenic seeds had higher germination rates, fresh weights and longer roots under salt stress compared with WT *Arabidopsis*, which was consistent with the results in several other crops including cotton and soybean. In the Salt Overly Sensitive pathway, the calcium-ion binding protein SOS3 senses and binds to the calcium signal caused by salt stress, and activates the Ser/threonine protein kinase SOS2, which phosphorylates the Ser residue in the DSPS unit at the C-terminal of SOS1 protein, then the functional active region of SOS1 protein is released, so that the $Na^+$ can be excreted from cells [11,35]. In this study, overexpression of *RsSOS1* improved salt tolerance of transgenic *Arabidopsis*, which may be due to the fact that radish RsSOS1 protein has a DSPS structure being similar to that of AtSOS1 protein. When transgenic Arabidopsis is subjected to salt stress, AtSOS2 was activated by AtSOS3, and then phosphorylated the DSPS region of exogenous RsSOS1 to activate its ion transport activity. The function of *RsSOS1* is very similar to that of the *SOS1* gene in other plants, suggesting that the contribution of *SOS1* to plant salt tolerance is conserved. *RsSOS1* mainly participates in the enhancement of $Na^+$ extrusion and the regulation of $Na^+$ and $K^+$ homeostasis in plant roots under salt stress, thus improving root vigor, reducing cell membrane damage and maintaining normal plant growth [40]. Plant salt tolerance is a quantitative trait controlled by multiple genes, involving a complex regulatory network of many signaling pathways. The salt stress response mechanism of plants is more complicated than in the single-celled organism yeast, which needs to be further studied.

More evidence has demonstrated that SOS1 with a short cut in the autoinhibitory C-terminal domain can confer yeast mutant AXT3 with strong salt tolerance [11]. It is speculated that radish *RsSOS1* also has a similar mechanism. In this study, homologous comparison was conducted between the amino acid sequence of radish RsSOS1 and AtSOS1, and it was inferred that the autoinhibitory region of RsSOS1 was located behind the 999th amino acid according to the known autoinhibitory domain of AtSOS1. Through the yeast functional complement analysis, it was found that the transferred *RsSOS1* could not compensate the defective function of $Na^+$ efflux of the yeast mutant AXT3, which was consistent with the results in *Bruguiera gymnorrhiza* [41]. While the *SOS1* of Arabidopsis, rice and wheat could partially compensate the defective function of yeast mutant AXT3 [9,11,42]. After removing the autoinhibitory region in *SOS1* of the above species, AXT3 yeast could recover growth with lower $Na^+$ content, higher $K^+$ content and a higher $K^+/Na^+$ ratio under

severe salt stress conditions. Similar results were obtained in this study. In conclusion, the $Na^+/H^+$ antiporter domain exists before the 999th amino acid of RsSOS1, which performs the function of sodium ion exportation and transport in cells under salt stress, laying a foundation for the further investigation of the functional domain of RsSOS1. When exogenous *RsSOS1* was transferred into AXT3, it was speculated that due to the lack of homologous genes to activate *RsSOS1* in yeast, the ion transport functional domain could not be released. However, the *RsSOS1△999* was in a hyperactive state without auto inhibitory region, so it could effectively compensate the defective function of AXT3 yeast. The RsSOS1 protein without the autoinhibitory region acts as a super-active ion transporter, and could be used as a new salt-tolerance gene in the genetic improvement of salt tolerance traits in crops. This result would facilitate clarifying the mechanisms underlying salt tolerance, and the breeding of elite salt-tolerant cultivars of radish.

**Supplementary Materials:** The following are available online at https://www.mdpi.com/article/10.3390/horticulturae7110458/s1, Figure S1. Isolation of *RsSOS1* gene. Figure S2. Predicted transmembrane domain of RsSOS1 protein. Table S1. Primers used in this study.

**Author Contributions:** Conceptualization, Y.Z. and L.L.; Data curation, J.D.; Formal analysis, Y.W., L.X., Y.Z. and L.L.; Funding acquisition, Y.W., L.X. and L.L.; Investigation, W.Z., J.L., K.L. and X.Y.; Methodology, W.Z., J.L., J.D., L.X., K.L., X.Y. and L.L.; Project administration, L.X., Y.Z. and L.L.; Supervision, Y.Z. and L.L.; Validation, J.L.; Visualization, W.Z. and J.D.; Writing—Original draft, W.Z.; Writing—Review and editing, W.Z., J.L., J.D., Y.W., Y.Z. and L.L. All authors have read and agreed to the published version of the manuscript.

**Funding:** This work was partially supported by grants from the Jiangsu Agricultural Science and Technology Innovation Fund (CX (20)3144), the National Natural Science Foundation of China (31601766), the earmarked fund for Jiangsu Agricultural Industry Technology System (JATS[2021]462) and the Project Funded by the Priority Academic Program Development of Jiangsu Higher Education Institutions (PAPD).

**Institutional Review Board Statement:** Not applicable.

**Informed Consent Statement:** Not applicable.

**Data Availability Statement:** Data are contained in the article.

**Acknowledgments:** Thank Ling Yu, from the College of Resources and Environmental Sciences of Nanjing Agricultural University for supplying AXT3 yeast.

**Conflicts of Interest:** The authors declare no conflict of interest.

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
