# Peer review of "RsSOS1 Responding to Salt Stress Might Be Involved in Regulating Salt Tolerance by Maintaining Na+ Homeostasis in Radish (Raphanus sativus L.)"

_horticulturae, doi:10.3390/horticulturae7110458_

Round 1

Reviewer 1 Report

The manuscript “RsSOS1 Responding to Salt Stress Might Involve in Regulating Salt-tolerance by Maintaining Na+ Homeostasis in Radish (Raphanus sativus L.)” by Ljubej, et al., who have isolated the RsSOS1 gene from radish genotype ‘NAU-TR17’ and tried to identify its overexpression under different salt stress treatments. This research could be valuable in agriculture in particular to understand the molecular mechanism underlying salt stress response in radish.

The manuscript is well organized and thoroughly described however, there are some minor comments that I feel, should be addressed.  

At the start of the introduction section, the authors should mention the agronomic importance of the radish crop in order to relate their hypothesis developed for this study. Also, at the end of this section, say something about future prospects of this experiment in order to facilitate the researchers/farmers in this sector.

The M&M section is well written and properly described. However, I feel there should be a separate heading for describing the statistical analysis.

Figure 3A needs scale bars as well. The same is the case for figure 5E

The results are described very well however, the discussion of results needs some attention. The authors should elaborate a little more about the reasons for the specific results obtained, supported with the latest literature, in order to support their arguments developed for this study.

To sum up, this manuscript can find interest to readers in this field. I enjoyed reading it. However, I feel the above-mentioned minor suggestions may be incorporated before its possible publication.

Good Luck!

Reviewer 2 Report

The manuscript entitled “RsSOS1 Responding to Salt Stress Might Involve in Regulating

Salt-tolerance by Maintaining Na + Homeostasis in Radish (Raphanus sativus L.)” by Zhang and collaborators, clones and functionally characterized a RsSOS1 gene encoding a Na + / H + cotransporter involved in salt resistance. Authors present an interesting investigation, since the radish is an important agricultural species, as well as, the identification of genes involved in resistance to salt is of great interest for agriculture. However, the Ms presents methodological flaws, which mean that the claims presented in the results are not sustained. The Ms needs to rethink several experiments, as well as perform some additional ones. In particular, this reviewer thinks that authors need to take the following points into account:

In the experiments related to the localization of RsSOS1 protein, the authors present the localization of RsSOS1 in tobacco cells when they do not describe it in the method section. Furthermore, the 35s :: GFP line has a similar localization as the 35s :: RsSOS1-GFP. In addition, the cell nucleus should be marked, for example with DAPI, as well as the cell walls, for example using calcofluor.

Furthermore, it is surprising that the authors generate a transgenic line of Arabidopsis thaliana with 35s :: RsSOS1-GFP, because they do not study the cellular location in roots of this transgenic plant. These experiments would give more information at the cellular level of the location of this cotransporter.

The phylogenetic experiment is poorly described in the methods. The authors do not show a list of the species used as well as the proteins of each of them. They do not describe which model of evolution they use, as well as other parameters such as the number of bootstrapping replicates, etc. Furthermore, the authors in these phylogeny experiments use only one of the homologues present in each species used. A quick search for the tomato gene that codes for this protein in the SolGenomics database has identified another homologous gene in the tomato genome (Solyc04g018100.2, Sodium / hydrogen exchanger Na + H + antiporter). Have the authors identified all the homologous genes present in the species used? You need to use a species to act as an outgroup.

The claims of the authors in the results of these experiments are not supported by the evidence they present. The experiment must be carried out again.

Yeast experiments are confusing, and not all the results indicated in the results are presented. For example, they do not show the growth curves of the different transgenic yeasts. These experiments must be reanalyzed and explained in a better way. If talking about differences in growth, the curves from these experiments have to be shown.

The authors obtained little information on the different A. thaliana lines generated.

For these reasons, the reviewer thinks that the MS does not meet the standards necessary for its publication in Horticulturae journal

Reviewer 3 Report

The authors describe isolation and characterization of a plasma membrane Na+/H+ antiporter SOS1 from radish, as well as its expression in Arabidopsis and functional complementation in yeast. The work is well conducted and provides new insights into the role of RsSOS1 in Na homeostasis regulation. I recommend this article for publication in Horticulture after addressing the following minor corrections, most of which are related to the incomplete and/or unclear Materials and Methods section.

02 – involve - change to: be involved

32 – The Introduction section should include the description of the SOS signal transduction pathway (which was very briefly described in Discussion), as well as main structural characteristics of SOS1 proteins from other plants, with emphasis on their C-terminal auto-inhibitory region. This is necessary for readers to understand the basis of experiments with yeast complementation.

67 – “isolated by RT-PCR with cDNA template” – template for RT-PCR is RNA, so please correct this and indicate tissue from which RNA was isolated, the method for RNA or mRNA isolation and the kit used for RT-PCR and type of RT primers.

71 – The pCAMBIA1300 vector does not feature GFP (see https://www.abcam.com/pcambia1300-plant-expression-vector-ab275754.html#lb ), so where does the GFP come from? Please provide sufficient details on the vector construction.

72-73 – “…transferred into E.coli DH5α with heat shock method.” –this method requires a reference.

73 - The positive clones identified by PCR… - which pair of primers from Table 1 was used?

79 – Table 1 – Please provide all details that would allow easier understanding of your experimental procedures, including a column which would indicate the purpose (experimental step) where the primers were used, the targets for these primers (and their accession numbers where applicable), the expected amplicon lengths etc. What are DNA1 and DNA2?  What are lowercase and uppercase letters used for? Correct RsSOS1-qRT-F/R to RsSOS1-qRT-R

88-92 – You did not mention preparing constructs with GFP only and using them for transient transformation of tobacco, but later you provide images with both constructs (Fig 2). Also, specify tobacco species that was used – N. tabacum or N. benthamiana?

90 - transient transformation – provide a reference for the protocol.

102 – Section 2.5 – Please provide the cycling program for qPCR. Have you checked the specificity of primers and how?

116-118 – “The positive transgenic plants were screened continuously on ½ MS medium containing 36 mg/L hygromycin to obtain the homozygous T3 generation plants.” Have you really grown 3 generations of Arabidopsis, from regenerated plantlets to seeds and then from seed to seed, all in tissue culture? And they flowered in vitro to allow successful crossings? Have you performed crossings at all? Or is it possible that hygromycin selection alone can obtain homozygous plants, even though the SOS1 gene is overexpressed and should be dominant? Finally, why did you need homozygous plants at all? Please explain all this.

117 – Provide full name for ½ MS medium

119-120 – what is pCAMBIA1300 vector “universal primer”, which is not listed in Table 1?  What exactly does it amplify and in combination with what other primer and how long is the expected amplicon?

128-132 – This section on “self-activity inhibited region of RsSOS1” or At SOS1 is unclear without any explanation or references provided. Please provide some background on this and appropriate references – it would be best if you include this in Introduction. Later (line 295), it is termed “auto-inhibited domain”, but I suppose it should be auto-inhibiting or auto-inhibitory domain? Please check this, and apply uniform terminology throughout the manuscript.

137 – Please provide info on vector pYES2 manufacturer

139 – Please provide a reference for lithium acetate method

145 - …growth condition of yeast was observed - delete “conditions”. The conditions are set, and only growth is observed.

186 – Correct salt tolerant into salt tolerance; In the figure legend describe the statistical evaluation used and what are the asterisks.

204 – I think you should delete “the same as follows”.

214 - and highly than – change to: but higher than

223 – Please indicate the (expected) amplicon sizes on all three gels

224 – Please specify the primers used for Fig 5A? Is it“pCAMBIA1300 vector universal primer”? Again, what does it amplify? I guess so, because RsSOS1-DNA1-F and R and RsSOS1-DNA2-F and R primers, whatever they are, would amplify 3223 bp and 3001 bp segments respectively when genomic sequence is a template, but here the transgene is acquired from cDNA, so only 1360 bp or 2164 bp amplicons would be obtained, respectively. Please clarify this.

240 – Figure 6 – please indicate the yeast cell dilutions

269 - Since the firstly identification of the Na+/H+ antiporter gene SOS1 in A. thaliana…..reference?

Supplement – please provide a legend for A – what are samples 1, 2 and 3? As for the figure B, I think that the sequence analysis should be included in the main text.

Reviewer 4 Report

This manuscript presents the data on the functional validation of plasma membrane antiporter in radish. The authors found that RsSOS1 mediated Na+ efflux in AXT3 yeast and its overexpression in Arabidopsis significantly improved the salt tolerance of transgenic plants, indicating that RsSOS1 might be involved in radish response to salinity. The experiment was well done and the data were presented completely. I would like to recommend that this manuscript be accepted for publication after addressing the minor modifications below:

1. The author found that the expression level of RsSOS1 ‘in salt tolerant genotype was significantly higher than that in salt sensitive one’ under 150 mmol/L NaCl treatment. I suggest that A reasonable explanation for the above interesting phenomenon should be added after the first paragraph on page 11 of Discussion.

2. Some minor formatting errors should be corrected in the full text. For examples:

1) Line 21 on page 1: ‘in cell membrane’ should be changed to ‘in plasma membrane’.

2) Line 25-26 on page 1: ‘could obviously replenish the deficient Na+ efflux function of…’ should be changed to ‘could rescue Na+ efflux function of…’.

3) Line 46, 54 on page 2: ‘Na+’ and ‘H+-ATPase’, ‘+’ should be superscript.

4) Line 66-67 on page 2: ‘Na+/H+ reverse transport’ should be changed to ‘Na+/H+ exchange transport’.

5) I suggest that Tables 1 on page 2 and Figure 1 on page 5 can be moved to Supplementary Materials additional materials.

6) Line 315 on page 11: ‘A. thaliana’ should be italicized.

Round 2

Reviewer 2 Report

The revised version of manuscript entitled “RsSOS1 Responding to Salt Stress Might Involve in Regulating Salt-tolerance by Maintaining Na + Homeostasis in Radish (Raphanus sativus L.)” by Zhang and collaborators, it has improved in terms of quality, however the authors have not solved several of the article's bad problems. The details that the authors have not addressed and are necessary for the publication of the article. Among them:

Phylogenetic relationships, to be valid, must be performed with a large number of sequences. To obtain valid results, it is necessary to work with orthologous genes, or failing that, homologous genes based on blast searches. The authors have different databases to obtain the orthologues of the A. thaliana genes used in the different species of the tree.

Here is an interesting reference on the subject:

Natsidis P, Kapli P, Schiffer PH, Telford MJ. Systematic errors in orthology inference and their effects on evolutionary analyses. iScience. 2021 Jan 28;24(2):102110. doi: 10.1016/j.isci.2021.102110. PMID: 33659875; PMCID: PMC7892920.

The phylogenetic tree has to be done again including more number of sequences, to check the plausibility of the results.

In addition, it is necessary to include the bootstraping values obtained in the phylogenetic analysis.

Redoing the experiment is necessary for the inclusion of these results in the publication.

The legend in figure 2 needs to be improved to make it more self-explanatory

It is necessary to include a group of sequences as an outgroup

The images of the co-location are of very low quality and it is difficult to conclude anything. In the references cited by the authors appear images of much higher quality and that if they allow to support the sublocation.

The images have to be redone.

Finally, why the authors do not want to show the results in A. thaliana transgenic lines?

In reference to transgenic yeast growth experiments, it is necessary to perform a growth curve. This must be included in the supplementary material.

The images shown in figure 7 do not support the claims about the different growth rates of the different yeast lines used by the authors.

For example

https://www.biotek.com/resources/application-notes/monitoring-growth-of-beer-brewing-strains-of-saccharomyces-cerevisiae-the-utility-of-synergy-h1-for-providing-high-quality-kinetic-data-for-yeast-growth-applications/

Lastly, the accession numbers MZ484951 and MZ484950 are not available on GenBank. It is necessary that these numbers are public to be able to verify what the authors argue.

What ration does the RsSOS1 gene have with the radish genome entry XM_018601403.

Regarding the cloning of the gene, it is necessary to analyze if it appears in the radish genome, as well as it is necessary to include in the supplementary materials a scheme of the gene, where all the elements (exons and introns) are indicated, as well as other structural elements.

In general, the authors have not answered the doubts of this reviewer, since they suggested experiments that the authors have not carried out.

For this reason, this reviewer maintains his opinion on rejecting the MS.

Author Response

Many thanks for the reviewer’s suggestions. In this study, our works mainly focus on characterization of RsSOS1, and SOS1 of other species were carried out to compare the genetic relationship of SOS1 from different species.
Furthermore, we are trying to carry out relevant investigation for further validating the subcellular localization of 35s :: RsSOS1-GFP protein in roots of transgenic plant in the near future.